# ACDiT: Interpolating Autoregressive Conditional Modeling and Diffusion Transformer

**Jinyi Hu**[1*]    **Shengding Hu**[1*]    **Yuxuan Song**[1]    **Yufei Huang**[1]    **Mingxuan Wang**[2]

**Hao Zhou**[1]    **Zhiyuan Liu**[1]    **Wei-Ying Ma**[1]    **Maosong Sun**[1†]

[1]*Tsinghua University*    [2]*ByteDance*

*{hu-jy21,hsd23}@mails.tsinghua.edu.cn*

**Reviewed on OpenReview:** *https://openreview.net/forum?id=OuFNXESoCO*

## Abstract

Autoregressive and diffusion models have achieved remarkable progress in language models and visual generation, respectively. We present ACDiT, a novel **A**utoregressive block-wise **C**onditional **Di**ffusion **T**ransformer, that innovatively combines autoregressive and diffusion paradigms. By introducing a block-wise autoregressive unit, ACDiT offers a flexible interpolation between token-wise autoregression and full-sequence diffusion, bypassing the limitations of discrete tokenization. The generation of each block is formulated as a conditional diffusion process, conditioned on prior blocks. ACDiT is easy to implement, as simple as applying a specially designed Skip-Causal Attention Mask on the standard diffusion transformer during training. During inference, the process iterates between diffusion denoising and autoregressive decoding that can make full use of KV-Cache. We validate the effectiveness of ACDiT on image, video, and text generation and show that ACDiT performs best among all autoregressive baselines on similar model scales on visual generation tasks. We also demonstrate that benefiting from autoregressive modeling, pretrained ACDiT can be transferred to visual understanding tasks despite being trained with the generative objective. The analysis of the trade-off between autoregressive and diffusion demonstrates the potential of ACDiT to be used in long-horizon visual generation tasks. We hope that ACDiT offers a novel perspective on visual autoregressive generation and sheds light on new avenues for unified models.

## 1 Introduction

Autoregressive modeling has been a central paradigm in artificial intelligence, most notably driving the success of large language models through next-token prediction (Touvron et al., 2023a;b; Brown, 2020; Achiam et al., 2023; Jiang et al., 2023; Bai et al., 2023; Yang et al., 2024). By decomposing complex distributions into a sequence of conditional predictions, autoregressive models provide a natural mechanism for modeling long-range dependencies and structured generation. Similar predictive formulations also underpin reinforcement learning (Mnih, 2013; Schulman et al., 2017; Chen et al., 2021; Reed et al., 2022) and world models (Ha & Schmidhuber, 2018), where future states are predicted conditioned on past observations. These successes motivate us to revisit autoregressive factorization beyond language and ask how its advantages can be effectively extended to high-dimensional visual generation.

In the realm of visual generation, diffusion models (Ho et al., 2020; Song et al., 2020a; Dhariwal & Nichol, 2021; Song et al., 2020b) have demonstrated superior generative capabilities, producing creative outputs that are

---

*Equal contribution

†Corresponding authors

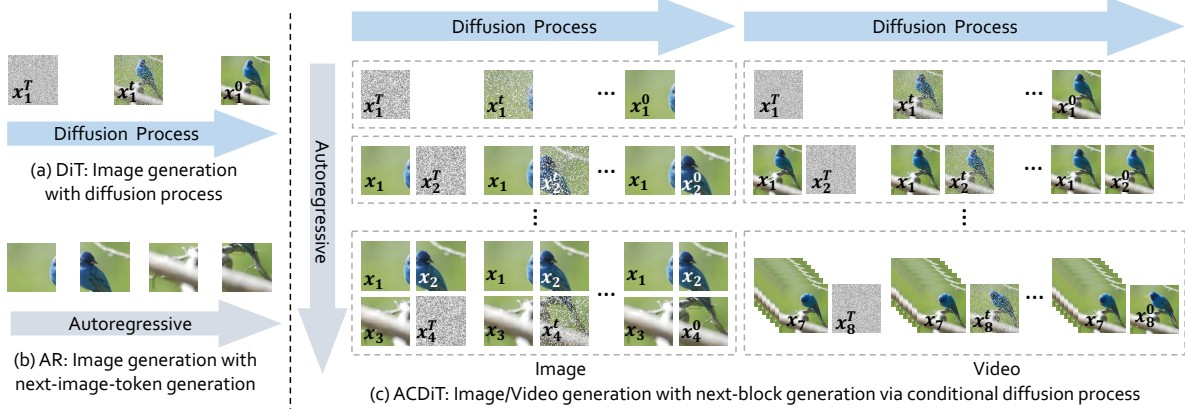

Figure 1: The generation process of ACDιT. Pixels in each block are denoised simultaneously conditioned on **clean context** which are autoregressively generated. The block can be flexibly defined, such as part of pixels in images or several frames in videos.

virtually indistinguishable from human-generated content, as evidenced by innovations like Sora (Brooks et al., 2024) and Stable Diffusion (Rombach et al., 2022; Podell et al., 2023; Esser et al.). Despite their remarkable success, diffusion models operate in a non-autoregressive manner. They take as input corrupted target sequences of the full length and reconstruct the intended output through *in-place* iterative refinement. The term "in-place" highlights a fundamental distinction from autoregressive models, which provide subsequent prediction, thereby extending the sequence and progressing towards the future. Such variation makes the diffusion model less effective at capturing long-range temporal dependencies in visual data and poses difficulties for developing integrated frameworks capable of seamlessly bridging the vision foundation model with unified multimodality modeling (Dong et al., 2024; Team, 2024; Zhou et al., 2024a; Wang et al., 2024b; Xie et al., 2024; Wu et al., 2024b) and world model (Yang et al., 2023; Du et al., 2023; Bruce et al., 2024; Zhou et al., 2024b; Wu et al., 2024a).

To build the visual autoregressive model, existing works convert visual generation tasks into discrete token prediction tasks with vector quantization techniques (Esser et al., 2021; Team, 2024; Wang et al., 2024b; Tian et al., 2024) and training with a next-token prediction objective. However, approaching the continuous distribution requires huge vocabulary sizes and a high utilization rate (Yu et al., 2023b; Weber et al., 2024), which is a complex objective.

In this paper, we propose ACDιT, an **A**utoregressive blockwise **C**onditional **Di**ffusion **T**ransformer that fuses the diffusion process with the autoregressive paradigm. Instead of relying on discrete tokens, we demonstrate that visual autoregressive generation can also be achieved using continuous visual features. At a high level, we extend the autoregressive units from the individual discrete token to blocks, where each block consists of visual patches of flexible size. The generation of each block can be formulated as a conditional diffusion process based on the previous block. As Fig. 1 shows, for image generation, a block represents a small region of the image, whereas in video generation, a block can correspond to a single frame or multiple frames.

ACDιT is easy to implement, as simple as adding a tailored Skip-Causal Attention Mask to the current diffusion transformer (Peebles & Xie, 2022) during training. The inference process is formatted as an iteration between the conditional diffusion denoising process within a block, conditioned on the complete clean context, and autoregressive generation of a new block appended as the new context. In this way, KV-Cache can be used for faster inference. In general, ACDιT offers the following inherent advantages: (i) ACDιT simultaneously learns the causal interdependence across visual blocks with autoregressive modeling and the non-causal dependence within blocks with diffusion modeling. (ii) ACDιT is endowed with clean and continuous visual input, which eliminates the requirement for vector quantization and improves the transfer to visual understanding tasks after being trained with generative objectives. (iii) ACDιT makes full use of KV-Cache for flexible autoregressive generation of any length and can potentially benefit from other latest techniques in text for long video generation.

The effectiveness of ACDiT is primarily validated on visual generation tasks, including both image and video generation tasks, respectively. To further demonstrate its versatility, we also perform experiments on text generation. Experimental results demonstrate that ACDiT outperforms all autoregressive baselines of the same model scale, and achieves visual quality comparable to full-sequence diffusion models while exhibiting higher inference speed when extended to long sequences. It also achieves strong performance on text generation following the discrete diffusion formulation without any specific customization. We hope ACDiT presents a fresh perspective on the visual autoregressive model and paves the way for exploring unified multimodal and world models. The codes and models are available at https://github.com/thunlp/ACDiT.

## 2 Related Work

**Diffusion Models.** The field of image generation has witnessed remarkable advancements with the introduction of diffusion models (Ho et al., 2020; Song et al., 2020a; Dhariwal & Nichol, 2021; Nichol & Dhariwal, 2021). U-Net (Ronneberger et al., 2015) is the early mainstream choice of network architecture (Song et al., 2020b; Nichol & Dhariwal, 2021; Rombach et al., 2022; Podell et al., 2023). Following that, Transformer (Vaswani et al., 2017) is applied to diffusion models for image generation, with groundbreaking work such as DiT (Peebles & Xie, 2022) and U-ViT (Bao et al., 2023). A series work, including PixArt-$\{\alpha, \delta, \Sigma\}$ (Chen et al., 2023; 2024c;b), demonstrate the capability of DiT on text-to-image tasks. Several studies also apply DiT to video generation, such as Lumiere (Bar-Tal et al., 2024) and Movie Gen (Polyak et al., 2024).

**Autoregressive Generation on Discrete Tokens.** The iGPT (Chen et al., 2020) first proposes autoregressively generating raw image pixels as a raster-scan sequence. Most visual autoregressive models (Ramesh et al., 2021; Wang et al., 2024b; Yu et al., 2022; Ding et al., 2021; 2022) follow VQGAN (Esser et al., 2021), which pioneers this direction by training an autoregressive transformer on discrete tokens produced by VQVAE (Van Den Oord et al., 2017). LlamaGen (Sun et al., 2024) enhances the image tokenizer and scales up the autoregressive transformers building on the latest Llama architecture (Touvron et al., 2023a). Inspired by RQ-Transformer (Lee et al., 2022), VAR (Tian et al., 2024) proposes the next-scale prediction and obtains good improvement. Some subsequent works (Tian et al., 2024; Li et al., 2024c; Yao et al., 2024) extend next-scale prediction to text-to-image generation.

**Autoregressive Generation on Continuous Features.** Some recent works make some early exploration on achieving visual autoregressive generation without discrete tokens and combining the advantages of diffusion and autoregressive. Diffusion Forcing (Chen et al., 2024a) trains a causal autoregressive model to generate blocks without fully diffusing past ones and implements it on small RNN. MAR (Li et al., 2024a) proposes the diffusion loss to learn the autoregressive conditional distribution on the head of the main Transformer with a small MLP network. Transfusion (Zhou et al., 2024a) and Monoformer (Zhao et al., 2024a) conduct joint training with language modeling loss and diffusion loss in a single transformer, while within image generation, their design is identical to the standard diffusion model. Dart (Gu et al., 2024) proposes to autoregressively denoise the complete image within in a Transformer. Different from these works, ACDiT redefines the autoregressive unit and generates each block based on the clear past. Block Diffusion (Arriola et al., 2025) also explores an interpolation between autoregressive and diffusion modeling. Their approach focuses solely on text generation, building on discrete diffusion objectives proposed in prior work (Sahoo et al., 2024; Shi et al., 2024), and introduces variable-length generation and KV-caching within diffusion language models. In contrast, ACDiT is designed for visual generation, and is motivated by enabling autoregressive modeling over continuous visual data, yet still achieves better results on text generation using our simple and generally effective model architecture.

## 3 Prerequisite

### 3.1 Autoregressive Modeling

Autoregression asserts that the value at each timestep is contingent upon its preceding values. This principle is exemplified in autoregressive language models, which iteratively predict the probability distribution of subsequent tokens. Given a sequence of tokens $(x_1, x_2, \ldots, x_n)$, a salient characteristic of autoregression is that the prediction of $x_i$ is only dependent on its prefix $(x_1, x_2, \ldots, x_{i-1})$. Upon determining $x_i$, it is

concatenated with the preceding sequence, thereby forming the conditioning context $(x_1, x_2, \ldots, x_i)$ to predict $x_{i+1}$. Therefore, the likelihood of sequence can be factorized as:

$$p(x_1, x_2, \ldots, x_N) = \prod_{i=1}^{N} p(x_i | x_{<i}).$$ (1)

Thanks to the flexibility of self-attention in Transformers, autoregressive models can be effectively implemented by adding a causal attention mask in the Transformer attention block (Vaswani et al., 2017).

### 3.2 Diffusion

Diffusion models, in contrast, conceptualize a noise-infusion and denoising process, which is defined by gradually adding noise to the initial data $x_0 \sim p(x)$ and training the model to learn the inverse mapping. Formally, the noised data $x^{(t)}$ at each step $t$ is sampled by $q(x^{(t)}|x^{(t-1)}) = \mathcal{N}(x^{(t)}; \sqrt{\alpha^{(t)}}x^{(0)}, (1-\alpha^{(t)})\mathbf{I})$[1], which is equivalent to add a Gaussian noise to the previous samples $x^{(t)} = \sqrt{\alpha^{(t)}}x^{(t-1)} + \sqrt{1-\alpha^{(t)}}\epsilon^{(t)}, \epsilon^{(t)} \sim \mathcal{N}(0, \mathbf{I})$. $p_\theta$ is trained to learn the reverse process $p_\theta(x^{(t-1)}|x^{(t)}) = \mathcal{N}(\mu_\theta(x^{(t)}), \beta^{(t)}\mathbf{I})$. With the reparameterization trick, the network $\mu_\theta(x^{(t)})$ can be reparameterized as noise prediction network $\epsilon_\theta(x^{(t)})$ and the training objective can be simple as:

$$\mathcal{L}_\theta = \mathbb{E}_{t \sim U[0,1], \epsilon \sim \mathcal{N}(0,I)} ||\epsilon_\theta(x^{(t)}, t) - \epsilon^{(t)}||_2.$$ (2)

During the inference phase, the denoising process is initialized with a random Gaussian noise sample $x^{(T)}$, followed by $T'$ denoising steps, ultimately yielding a single deterministic sample $\tilde{n}^{(0)}$ from its underlying distribution. Typically, the denoising process in diffusion models operates "in-place", meaning that each new denoising step directly replaces the previous step's input. This differs from autoregressive modeling, where the value of a subsequent step is appended to the existing sequence.

## 4 ACDiT

### 4.1 Desiderata for Autoregressive Diffusion Model

Rather than modeling the full data distribution $p(x)$ directly, autoregressive diffusion aims to learn the conditional distributions for the continuous visual feature $p(x_i|x_{<i})$ through a diffusion process, which is typically represented as categorical distributions over discrete token vocabulary in language models. Combining the Eq.(1) and Eq.(2), the learning objective of autoregressive diffusion is:

$$\mathcal{L}_\theta = \mathbb{E}_{t \sim U[0,1], \epsilon \sim \mathcal{N}(0,I)} \sum_{i=1}^{N} ||\epsilon_\theta(x_i^{(t)}; t, x_{<i}) - \epsilon^{(t)}||_2,$$ (3)

where $\{x_1, x_2, ..., x_N\}$ are $N$ autoregressive units. In the context of visual generation, each unit can flexibly correspond to the continuous visual information of a patch of pixels in an image or a set of frames in a video, depending on the granularity of the representation.

The key difference between Eq.(3) and Eq.(2) is that for each input noisy block, the noise prediction network $\epsilon(x_i^{(t)})$ predicts the noise conditioned on their previous clean block. To effectively learn this objective, we need a framework that integrates the strengths of both autoregressive and diffusion models. We identify three critical desiderata the framework should meet:

1. *The generation of future elements should be predicated on a precise representation of antecedent sequences.* This is imperative because any ambiguity in the past inevitably complicates future predictions. This approach preserves the efficacy of autoregressive modeling and potentially facilitates the development of the world model. Furthermore, adherence to this principle enhances performance in discriminative tasks (e.g., visual understanding), as these tasks necessitate the input of all observable features into the model.

---

[1]For clarification, we use subscript $_t$ to denote timesteps in autoregressive models and superscript $^{(t)}$ to denote the timesteps in diffusion models.

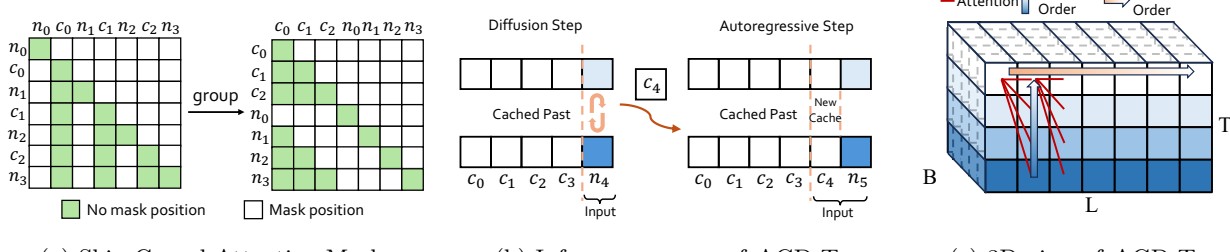

(a) Skip-Causal Attention Mask.      (b) Inference process of ACDɪT.      (c) 3D view of ACDɪT.

Figure 2: **(a)**: For each noised block $n_i$, it can only attend previous clean block $\{c_0, c_1, \ldots, c_{i-1}\}$ and itself. Each clean block $c_i$ only attends to the previous clean blocks. **(b)**: ACDɪT can effectively utilize the KV-Cache during inference. **(c)**: the 3D view of ACDɪT, where $B, L, T$ denote the block size, number of blocks, and denoising time steps, respectively. The darker color denotes higher noise.

2. *Both the autoregressive modeling and the denoising process should optimally utilize the entire parameter space of the neural network.* In an elegant fusion of autoregressive models and diffusion, neither component should be relegated to an auxiliary role. Instead, they should function as integral and complementary elements of the system.

3. *The denoising process should directly attend comprehensively to the entire sequence of past sequences.* Otherwise, the conditioning input to the denoising process would be required to fully encapsulate all prior information, imposing an unrealistic burden on the bottleneck context vector to losslessly compress the context feature into one vector. A holistic input of all past information in each denoise step ensures a more effective processing of temporal dependencies.

Based on these desiderata, we analyze representative autoregressive diffusion methods. **Diffusion-Forcing** (Chen et al., 2024a) proposes to use different-level random noise in different positions of a sequence. Thus, in the inference process, denoising subsequent positions from a clean past can be seen as a special case of denoising different levels of noise. Their method does not meet the first desideratum. In the training process, the future is not predicted from the precise representation of the past. In **MAR** (Li et al., 2024a), the diffusion process is trained based on the block in the last position, which does not satisfy the third desideratum. Moreover, the diffusion process sorely leverages the head part of the network, which conflicts with the second desideratum. To address suboptimal performance due to the causal direction based on this design, MAR incorporates bidirectional attention, which requires recomputing attention at each step and prevents the use of KV-cache during inference. In **Transfusion** (Zhou et al., 2024a) and **Monoformer** (Zhao et al., 2024a), the autoregressive generation is restricted to the textual modality and is not applied within the visual modality. Image generation in these models follows the standard diffusion process conditioned on preceding text inputs, without incorporating autoregressive prediction over visual patches. Moreover, in visual understanding tasks or multi-image generation settings, subsequent text and images attend to the noisy version of the preceding image, which conflicts with the first desideratum. To mitigate this issue, these models reduce the noise schedule, i.e., halving the maximum number of diffusion steps in 20% of image-captioning pairs, which is not an optimal strategy. See the visual comparison of these models in Fig. 3a.

## 4.2 Framework

To satisfy the desiderata discussed above, we propose a versatile framework for autoregressive diffusion called ACDɪT. For generality, ACDɪT runs block-wise autoregression instead of token-wise autoregression. We identify two kinds of blocks $c_i$ and $n_i$. For each autoregressive unit $x_i$, $c_i$ corresponds to the clean version $x_i$, $n_i$ corresponds to the corrupted version $x_i^{(t)}$. ACDɪT learns the conditional noise prediction network $\epsilon_\theta(n_i^{(t)}; t, c_{<i})$ with Eq.( 3).

To effectively learn this objective within one transformer network, we can conceptualize all $n_i$ and all $c_i$ as separate positions, effectively transforming the dependency structure into an attention pattern between different positions. We designate this attention pattern as the Skip-Causal Attention Mask, shown in Figure 2a.

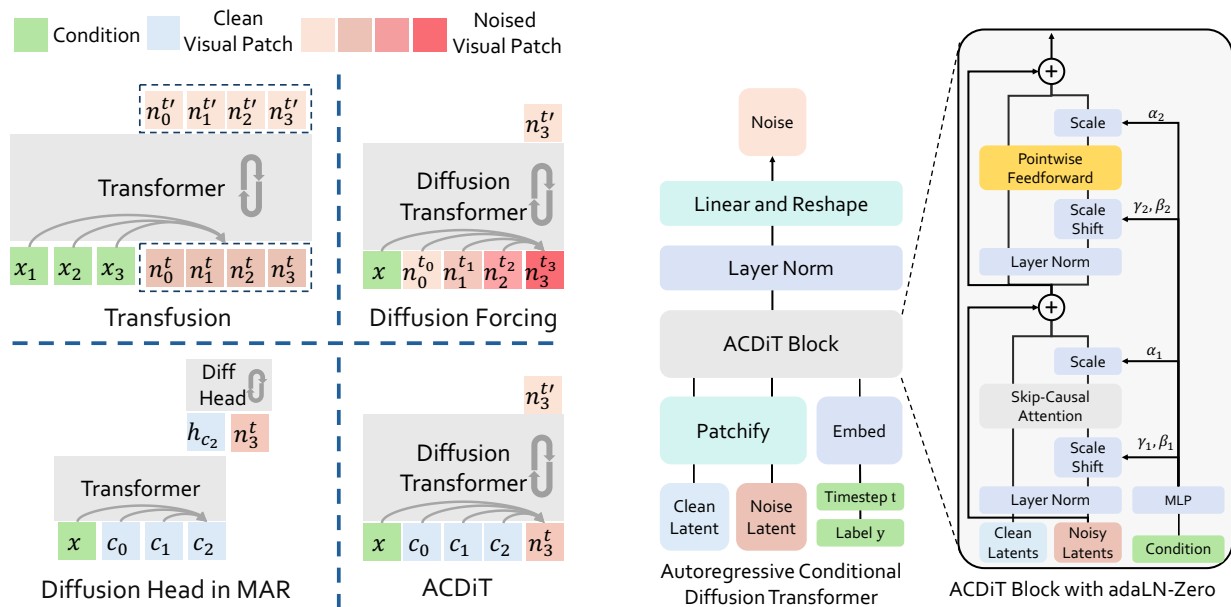

(a) Comparison of model design.

(b) The model architecture of ACDiT.

Figure 3: **(a)**: Comparison between ACDiT with Transfusion (Zhou et al., 2024a), Diffusion Forcing (Chen et al., 2024a) and MAR (Li et al., 2024a).Transfusion does not perform autoregressive modeling within the visual modality. Diffusion Forcing introduces different noise levels across the autoregressive inputs. MAR applies diffusion only on top of the backbone model. ACDiT utilizes the full parameters for both the autoregressive and diffusion process with clean input as context. **(b)**: The model architecture of ACDiT. Both the clean and noisy latent are input, while only the noisy latent interacts with the conditioning information. The Skip-Causal Attention Mask pattern, shown in Figure 2a, will be applied in the attention block.

The figure elucidates that $n_i$ attends to all preceding clean blocks $\{c_j | j = 0, ..., i-1\}$ and itself, while $c_i$ also attends to all preceding clean blocks $\{c_j | j = 0, ..., i-1\}$ and itself. In training, for simplicity, we can group attention mechanisms as illustrated in the right matrix of Figure 2a. Suppose the number of blocks is $N$, then the unmasked positions form two triangular matrices with length $N-1$, complemented by a diagonal matrix with length $N$.

During the inference phase, each autoregressive step executes a conditional diffusion process for $n_i$ based on $\{c_0, c_1, ..., c_{i-1}\}$'s KV-Cache. Upon finishing denoising, it is appended to the clean sequence as $c_i$ followed by the maximal noisy version of the next block $n_{i+1}$. The key-value tensor will be computed for these two blocks, and the key-value tensor of the clean block $c_i$ will be kept in KV-Cache. All noisy version of $n_i$ is disregarded. The process is visualized in Figure 2b. A three-dimensional view is presented in Figure 2c. By interpolating between full-sequence diffusion and autoregressive paradigms, ACDiT enjoys flexibility and expressivity, enabling it to generate video of any length utilizing the latest long-context techniques developed for language models.

## 4.3 Positional Encoding

ACDiT is designed to be versatile, capable of handling one-, two-, three-, or even higher-dimensional data, including but not limited to text (1D), images (2D), and video (3D). For any given dimension of data, the position of that data is a critical attribute that must be made known to the model. This positional awareness enables the model to contextualize its current focus relative to historical data. In the domain of textual data, the Rotary Position Embedding (RoPE) (Su et al., 2024) has gained widespread adoption as an effective relative positional encoding method. To address the challenges posed by multi-dimensional positional indices, we introduce RoPE-ND, a natural extension of RoPE.

For a token of a $D$ dimensional data, its positional index is $[m_1, m_2, ..., m_D]$. Given query and key vectors in the Transformer's attention module, we partition the hidden dimension into $D$ segments. It is imperative that each segment's hidden dimension be an even number. For each segment $j$, we apply a RoPE with a specific base $b_j$, as defined in the following equation:

$$\mathbf{R} = \begin{bmatrix} \mathbf{R}^{d_1}_{\Theta_1, m_1} & 0 & \cdots & 0 \\ 0 & \mathbf{R}^{d_2}_{\Theta_2, m_2} & \cdots & 0 \\ \vdots & \vdots & \ddots & \vdots \\ 0 & 0 & \cdots & \mathbf{R}^{d_D}_{\Theta_D, m_D} \end{bmatrix},$$ (4)

where each $\mathbf{R}^{d_j}_{\Theta_j, m_j}$ represents a $d_j$-dimensional rotary matrix[2] with rotation angles $\Theta_j = \{b_j^{-2(i-1)/d_j}, i \in [1, 2, \cdots, \frac{d_j}{2}]\}$. The base $b_j$ is empirically determined as $100\lceil \frac{8L_j}{100\pi} \rceil$, where $L_j$ denotes the maximum position index in dimension $j$. This formulation ensures that the highest wavelength of RoPE is approximately eight times the maximum position, thereby mitigating rapid decay in long-term dependencies. It is worth noting that ACDIT inherently supports length extrapolation (Su et al., 2024), although a comprehensive exploration of this falls beyond the present work.

### 4.4 Efficiency Analysis and Block Size Choice

We provide a brief analysis of the computational efficiency of ACDIT in terms of floating-point operations (FLOPS). The full derivation and more analysis are discussed in the Appendix A. We first assume that denoising each block requires the same number of time steps $T$ as the full sequence diffusion, despite that when the block size is small, it may require fewer denoising steps, thus making ACDIT potentially more efficient. Let $L$ denote the sequence length, $h$ the hidden dimension, $\theta$ the number of parameters in a transformer layer, and $B$ the block size used in ACDIT. The FLOPS for standard full-sequence diffusion is $F_{\text{full}} = 2L\theta + 4hL^2$. With KV-Cache, ACDIT reduces the attention cost to $2hL^2 + 2hLB$. The relative FLOPS savings is: $\frac{F_{\text{saved}}}{F_{\text{full}}} = \frac{1 - \frac{B}{L}}{2 + \frac{\theta}{hL}}$. This indicates that ACDIT can reduce per-layer FLOPS by up to 50% when $L \gg B$.

### 4.5 Model Architecture and Implementation

As shown in Fig. 3b, ACDIT mainly inherits the main architecture of DiT. We replace the original bidirectional attention with the proposed SCAM attention pattern to process the clean and noisy latent. In addition, we replace the absolute position embedding and Layer Normalization with RoPE (Su et al., 2024) and RMSNorm (Touvron et al., 2023a), respectively. We use QK-norms for stable training. The additional condition, including timesteps and labels, is injected into the model with AdaLN-Zero only on the noise part.

We explore 4 different model sizes, as shown in Table 5. In the image generation task, we set the patch size as 1 and the autoregressive unit block size as $256 = 16 \times 16$. ACDIT is trained on ImageNet for 1.2M iterations with a batch size of 1024. We use the AdamW optimizer (Loshchilov, 2017) and WSD (Warmup Steady Decay) learning rate scheduling (Hu et al., 2024). We sample images with DPM-Solver (Lu et al., 2022) for 25 steps within each block and use classifier-free guidance (Ho & Salimans, 2022) with a guidance scale of 1.5. In video generation, we sample 16 frames from each video and set the patch size as 2 and the block size as $1024 = 256 \times 4$. We train ACDIT on UCF-101 for 400K iterations with a batch size of 96. The classifier-free guidance scale is 2.5. Unless otherwise specified, ablation and analysis experiments are conducted on ACDIT-B. See more details at Appendix 4.5.

## 5 Experiments

We mainly experiment on the ImageNet (Russakovsky et al., 2015) dataset with $256 \times 256$ resolution and the UCF-101 (Soomro et al., 2012) dataset with 16 frames for image generation and video generation, respectively. In addition, we also validate the generality of ACDIT on text generation.

---

[2]See a detailed explanation in the Equation 15 in (Su et al., 2024).

Table 1: Image generation results on ImageNet 256x256. We report FID score, Inception Score (IS), Precision (Pre.), and Recall (Rec.).

| Model | Type | Latent | KV-Cache | Params | FID↓ | IS↑ | Pre.↑ | Rec.↑ |
|---|---|---|---|---|---|---|---|---|
| ADM (Dhariwal & Nichol, 2021) | Diff. | - | - | 554M | 10.94 | 101.0 | 0.69 | 0.63 |
| LDM-4-G (Rombach et al., 2022) | Diff. | Cont. | - | 400M | 3.60 | 247.7 | - | - |
| DiT-L/2 (Peebles & Xie, 2022) | Diff. | Cont. | - | 458M | 5.02 | 167.2 | 0.75 | 0.57 |
| DiT-XL/2 (Peebles & Xie, 2022) | Diff. | Cont. | - | 675M | 2.27 | 278.2 | 0.83 | 0.57 |
| MaskGIT (Weber et al., 2024) | Mask. | Disc. | - | 227M | 6.18 | 316.2 | 0.83 | 0.58 |
| MAGE (Li et al., 2023) | Mask. | Disc. | - | 230M | 6.93 | 195.8 | - | - |
| VQGAN (Esser et al., 2021) | AR | Disc. | ✓ | 1.4B | 15.78 | 78.3 | - | - |
| RQTran (Lee et al., 2022) | AR | Disc. | ✓ | 3.8B | 7.55 | 134.0 | - | - |
| VAR-d16 (Tian et al., 2024) | VAR | Disc. | ✓ | 310M | 3.30 | 274.4 | 0.84 | 0.51 |
| VAR-d20 (Tian et al., 2024) | VAR | Disc. | ✓ | 600M | 2.57 | 302.6 | 0.83 | 0.56 |
| LlamaGen-L (Sun et al., 2024) | AR | Disc. | ✓ | 343M | 3.07 | 256.1 | 0.83 | 0.52 |
| LlamaGen-XL (Sun et al., 2024) | AR | Disc. | ✓ | 775M | 2.62 | 244.1 | 0.80 | 0.57 |
| LlamaGen-XXL (Sun et al., 2024) | AR | Disc. | ✓ | 1.4B | 2.34 | 253.9 | 0.80 | 0.59 |
| ImageFolder (Li et al., 2024b) | AR | Disc. | ✓ | 362M | 2.60 | 295.0 | 0.75 | 0.63 |
| LlamaGen+NPP (Pang et al., 2024) | AR | Cont. | ✓ | 1.4B | 2.55 | 282.0 | 0.84 | 0.56 |
| MAR-L-Causal (Li et al., 2024a) | AR | Cont. | ✓ | 479M | 4.07 | 232.4 | - | - |
| MAR-L (Li et al., 2024a) | Mask. | Cont. | - | 479M | 1.78 | 296.0 | 0.81 | 0.60 |
| MonoFormer (Zhao et al., 2024a) | AR+Diff | Cont. | - | 1.1B | 2.57 | 272.6 | 0.84 | 0.56 |
| Dart (Gu et al., 2025) | AR+Diff | Cont. | ✓ | 820M | 3.82 | 263.8 | - | - |
| ACDɪT-L | AR+Diff | Cont. | ✓ | 460M | 2.53 | 262.9 | 0.82 | 0.55 |
| ACDɪT-XL | AR+Diff | Cont. | ✓ | 677M | 2.45 | 267.4 | 0.82 | 0.57 |
| ACDɪT-H | AR+Diff | Cont. | ✓ | 954M | 2.37 | 273.3 | 0.82 | 0.57 |

## 5.1 Main Results

**Image Generation.** We report the FID-50K (Heusel et al., 2017), Inception Score (Salimans et al., 2016), Precision and Recall (Kynkäänniemi et al., 2019) of ACDɪT and baselines in Table 1. Compared with previous autoregressive models and masked generative models utilizing discrete tokens, such as VQGAN, VAR, LlamaGen, and MaskGIT, ACDɪT consistently achieves superior performance with lower FID scores at comparable model scales. Notably, ACDɪT-XL achieves 2.45 FID scores, outperforming both LlamaGen-XXL and VAR-d20 with similar parameters. Additionally, when compared to the MAR-L-Causal **variant that does not recompute attention**, ACDɪT-L significantly improves performance across all metrics, specifically improving FID from 4.07 to 2.53. Compared with other autoregressive diffusion methods, such as Monoformer (Zhao et al., 2024a) and Dart (Gu et al., 2024), ACDɪT has a significantly superior performance. When compared with leading diffusion-based methods, ACDɪT also demonstrates competitive performance. For instance, despite not employing full-sequence attention, ACDɪT models achieve results close to DiT-XL. In general, these results highlight the distinct advantages of ACDɪT over other baselines with the continuous latent representation and KV-Cache. Qualitative results are presented in the Appendix E.

**Video Generation.** Different from image generation, video inherently includes a temporal dimension, making it more well-suited to autoregressive modeling. The FVD metric on UCF-101 for class-conditional video generation is reported in Table 2. With hybrid AR+Diff architecture, ACDɪT-H achieves much lower FVD than other diffusion-based and autoregressive methods, even outperforming MAGVIT-AR and MAGVITv2-AR, which utilize a closed-source, specially designed video tokenizer. In contrast, ACDɪT simplifies the process by directly using an open-sourced image VAE. Although MAGVIT with masked generative methods has a lower FVD than ACDɪT, they rely on in-place operation to generate a video similar to the diffusion model and require repeatedly processing global context during inference. This constraint limits their ability and efficiency to generalize to long video generation and build world models. We further analyze the efficiency implications of autoregressive versus masked generative formulations in the following section, with quantitative inference-time comparisons reported in Table 6. Compared to image generation,

Table 2: Video generation results on the UCF-101 dataset. ACDɪT-XL-LT means ACDɪT-XL trained for longer epochs.

| Model | Type | Params | FVD↓ |
|---|---|---|---|
| LVDM (He et al., 2022b) | Diff. | 437M | 372 |
| Latte (Ma et al., 2024) | Diff. | 674M | 478 |
| Video-LaVIT (Jin et al., 2024) | Diff. | 7B | 281 |
| MagDiff (Zhao et al., 2024b) | Diff. | 2B | 340 |
| Matten (Gao et al., 2024) | Diff. | 853M | 211 |
| VideoFusion (Luo et al., 2023) | Diff. | 510M | 173 |
| MMVG (Fu et al., 2023) | Mask. | 230M | 328 |
| MAGVIT (Yu et al., 2023a) | Mask. | 306M | 76 |
| MAGVITv2 (Yu et al., 2023b) | Mask. | 307M | 58 |
| TATS (Ge et al., 2022) | AR | 331M | 332 |
| CogVideo (Hong et al., 2022) | AR | 9.4B | 626 |
| MAGVIT-AR (Yu et al., 2023a) | AR | 306M | 265 |
| MAGVITv2-AR (Yu et al., 2023b) | AR | 307M | 109 |
| OmniTokenizer (Wang et al., 2024a) | AR | 650M | 191 |
| ACDɪT-XL | AR+Diff. | 677M | 111 |
| ACDɪT-H | AR+Diff. | 954M | 104 |
| ACDɪT-H-LT | AR+Diff. | 954M | 90 |

Table 3: Classification accuracy on ImageNet.

| Model | Type | Params | Top-1 Acc |
|---|---|---|---|
| ViT-H (Dosovitskiy, 2020) | Supervised | 632M | 83.1 |
| MAGE (Li et al., 2023) | Masked | 328M | 84.3 |
| MAE (He et al., 2022a) | Masked | 632M | 85.9 |
| iGPT (Chen et al., 2020) | Generative | 1.4B | 72.6 |
| DiT-XL (Peebles & Xie, 2022) | Generative | 675M | 82.8 |
| ACDɪT-XL | Generative | 677M | 84.0 |

Table 4: Test perplexities on OpenWebText.

| | | PPL (↓) |
|---|---|---|
| **Autoregressive** | | |
| AR (Sahoo et al., 2024) | | 17.54 |
| **Diffusion** | | |
| SEDD (Lou et al., 2023) | | ≤ 24.10 |
| MDLM (Sahoo et al., 2024) | | ≤ 22.98 |
| **Block-wise diffusion** | | |
| BD3-LMs (Arriola et al., 2025) | $L' = 16$ | ≤ 22.27 |
| | $L' = 8$ | ≤ 21.68 |
| ACDɪT | $L' = 256$ | ≤ 22.98 |
| | $L' = 128$ | ≤ 22.82 |
| | $L' = 8$ | ≤ 21.59 |

ACDɪT demonstrates a greater potential in the more complex domain of video generation, where longer visual sequences and temporal information are critical. Qualitative results are presented in the Appendix E.

**Text Generation.** ACDɪT is a general, modality-agnostic framework. At the architectural level, ACDɪT adopts a unified block-wise autoregressive formulation and a shared attention pattern that conditions each block on clean past context while allowing non-causal modeling within the current block. This design is independent of the data modality and remains unchanged when applied to text generation. For text generation, which operates on discrete tokens, the differences from visual generation arise only in the noise process used during training and inference. Specifically, we follow discrete diffusion language modeling objectives and the same experimental setup as previous work (Sahoo et al., 2024; Shi et al., 2024). Importantly, these choices affect only the modality-specific corruption and denoising operators, while the model architecture, attention mask, and block-wise autoregressive structure remain identical. Specifically, we use a 12-layer Transformer with a hidden dimension of 768, 12 attention heads, and a 128-dimensional timestep embedding as our model backbone. We conduct all training and validation on the OpenWebText dataset (Gokaslan & Cohen, 2019), an open-source replica of the WebText dataset. To evaluate the model's ability in probabilistic modeling and

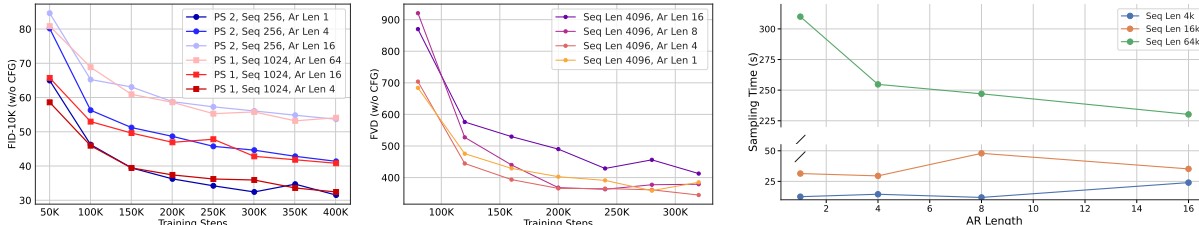

Figure 4: FID and FVD curves of ACDiT-B over training steps with different sequence lengths and autoregressive lengths. PS means patch size.

Figure 5: The change of inference time in different autoregressive lengths under varied total sequence length.

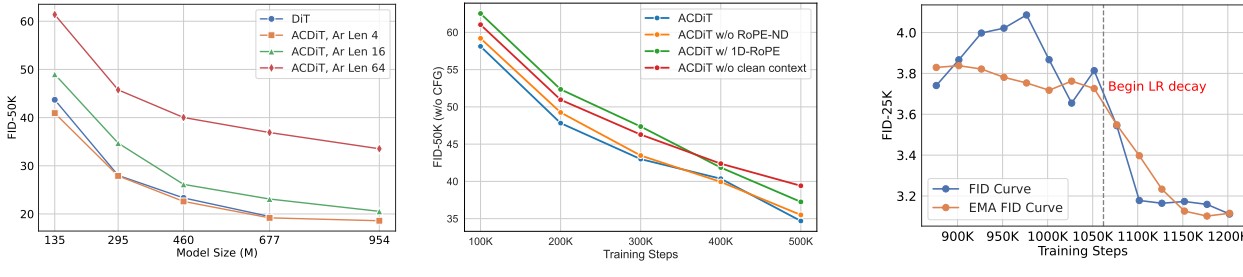

(a) Scaling performance of ACDiT.      (b) Ablation for ROPE-ND and SCAM.      (c) FID curve of last 30% training.

Figure 6: **(a)**: ACDiT shows scaling performance similar to DiT. **(b)**: Ablation for ROPE-ND and using clean contexts. **(c)**: FID score sharply drops with the learning rate decaying when using the WSD scheduler.

compression, we report Perplexity (**PPL**) on the validation set. PPL is defined as:

$$\text{PPL} = \exp\left(\frac{\mathbb{E}_{\boldsymbol{x}_0 \sim p_{\text{data}}}\left[-\log p_\theta(\boldsymbol{x}_0)\right]}{D}\right),$$

where $D$ is the data dimension. For models where the exact likelihood $p_\theta(\boldsymbol{x}_0)$ is intractable, we report the PPL calculated using the variational lower bound on the log-likelihood. Despite conceptual similarity to blockwise discrete approaches (Arriola et al., 2025), our effective positional embedding implementation (e.g., RoPE-ND; Sec. 4.3) enables ACDiT to achieve comparable performance across various block sizes as shown in Tab. 4 without any customized variance reduction or noise schedules as proposed in (Arriola et al., 2025). These advantages highlight the potential of ACDiT as a general and powerful generative modeling framework.

**Image Representation.** We assess the capability of ACDiT in image representation, which is essential for building a unified visual understanding and generation model. We finetune ACDiT-XL and DiT-XL on ImageNet using classification loss for 100 epochs and report the Top-1 accuracy in Table 3. ACDiT outperforms DiT-XL, highlighting the advantage of using clean latents, which help the model to learn better representations compared to using only noisy latent inputs. Additionally, ACDiT matches the Top-1 accuracy of MAGE (Li et al., 2023), while offering superior generation performance.

## 5.2 Analysis

**Trade off of block size.** Fig. 4 illustrates the trend of trade-off under different sequence lengths and block sizes in image and video generation tasks on ACDiT-B. The FID curve indicates that for image generation, directly increasing the autoregressive length leads to a decline in image quality, since each patch receives less attention information on average. However, we can mitigate this decline by increasing the total sequence length, which means reducing the patch size. For video generation, ACDiT shows more advantages due to the inherent temporal dependence of videos. The FVD curve demonstrates that increasing autoregressive length has minimal effect on the video quality, even with a slight improvement. As for efficiency, we test the sampling time for various sequence lengths with a batch size of 4 on an NVIDIA A100 GPU. Fig. 5 shows that as the sequence length increases, particularly beyond 16k, full-sequence attention (AR length of 1)

Table 5: Configuration details of ACDiT.

| Model | #Layers | Hidden | MLP | #Heads | Params |
|---|---|---|---|---|---|
| ACDiT-B | 12 | 768 | 3072 | 12 | 132M |
| ACDiT-L | 24 | 1024 | 4096 | 16 | 460M |
| ACDiT-XL | 28 | 1152 | 4608 | 18 | 677M |
| ACDiT-H | 32 | 1280 | 5120 | 20 | 954M |

Table 6: Inference cost comparison with MAR.

| Model | Params | Image | Video |
|---|---|---|---|
| MAR-B | 322M | 2.74s | 49.27s |
| MAR-L | 479M | 3.58s | 63.82s |
| ACDiT-L | 460M | 2.29s | 4.08s |
| ACDiT-XL | 677M | 3.22s | 5.16s |

becomes very time-consuming, necessitating the autoregressive generation. Empirically, blocks spanning 2–4 frames provide a good quality–efficiency trade-off for video generation. Dynamic or content-adaptive block sizing is a promising extension, and we leave it for future work.

**Inference Cost Comparison with MAR.** We further compare the average inference time of ACDiT and MAR for image generation and video generation in Table 6. Since MAR is not designed or trained for video generation, both models are evaluated under a controlled setting with a fixed sequence length of 4096 tokens to reflect inference cost in a long-horizon setting. Notably, ACDiT is consistently faster than MAR even for standard image generation. This shows that KV-cached block-wise autoregression already improves efficiency for common image-generation workloads. The advantage becomes substantially larger for long sequences. While MAR recomputes full bidirectional attention at each generation step, ACDiT caches key–value pairs of clean past blocks and only performs attention on newly generated blocks, avoiding redundant computation and achieving much lower (10x) latency for long-horizon video generation.

**Scaling Performance.** We present the scaling performance of ACDiT in Fig. 6a. For a fair comparison with DiT, we use the same batch size and learning rate as DiT in these training sessions. When increasing the model size, ACDiT shows consistent improvement in image quality across all autoregressive lengths, sharing a similar scaling trend with DiT. Notably, the performance gap between different autoregressive lengths shrinks as model capacity grows. This suggests that larger models can more accurately fit each autoregressive conditional distribution, thereby reducing the accumulation of errors over long autoregressive horizons.

**ROPE-ND.** We ablate the effectiveness of ROPE-ND on image generation. As shown in Fig 6b, both removing ROPE-ND and replacing RoPE-ND with standard 1D-RoPE consistently degrade FID.

**Effect of clean-past conditioning.** We further ablate the effect of clean-past conditioning by replacing clean context latents with noisy ones when modeling autoregressive conditionals. As shown in Fig. 6b, removing clean-past conditioning leads to consistently worse generation quality. Beyond generation, using clean latents also provides benefits for representation learning, as evidenced by the improved downstream classification performance discussed in Table 3. These results indicate that clean-past conditioning is beneficial for both autoregressive generation and learning informative visual representations.

**Training dynamics of WSD scheduler.** We utilize the WSD learning rate scheduler (Hu et al., 2024). WSD scheduler uses a constant learning rate during main training, while allowing divergence at any point with a rapidly decaying learning rate based on compute budget. As Fig. 6c shows, the FID remains almost converged during the constant learning rate state, while sharply dropping after the learning decays, similar to the loss curve when using the WSD scheduler in LLM training. To the best of our knowledge, we are the first to validate the effectiveness of the WSD scheduler in visual generation.

## 6 Conclusion

In this paper, we propose ACDiT that interpolates the autoregressive modeling and diffusion transformers. With a simple but novel design of attention mask, ACDiT can achieve autoregressive generation on any length while maintaining a clear latent input potentially for adding a visual understanding task. By combining the advantages of both autoregressive and diffusion models, we demonstrate the performance and efficiency of ACDiT in both image and video generation tasks. We hope ACDiT can shed light on designing the new visual autoregressive model and building a unified multimodal model in the future.

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

## A    Detailed Discussion about Block Size

In this section, we provide a detailed analysis of the computational cost of ACDiT in terms of FLOPS, comparing it to standard full-sequence diffusion in transformer layers. Let $L$ denote the sequence length, $h$ the hidden size, $n$ the number of attention heads (assumed $n \ll h$), $\theta$ the number of parameters in a single transformer layer, and $B$ the block size used in ACDiT.

**Full-Sequence FLOPS.**    Each transformer layer has two major computational components:

- Linear projections and feed-forward layers: $2L\theta$

- Q–K attention dot-products: $4(h+n)L^2 \approx 4hL^2$

Hence, the total FLOPS per layer is:
$$F_{\text{full}} = 2L\theta + 4hL^2$$

**ACDiT with KV-Cache.**    In ACDiT, attention is computed block-wise using KV-Cache. The $i$-th block (of size $B$) attends to $(i-1)B$ cached tokens, requiring $iB^2$ Q–K dot-products. Summing over all $\frac{L}{B}$ blocks:

$$\sum_{i=1}^{L/B} iB^2 = B^2 \cdot \frac{L}{B} \cdot \left( \frac{L}{B} + 1 \right) \bigg/ 2 = \frac{L^2 + LB}{2}$$

So the attention FLOPS becomes:

$$\text{Attention}_{\text{ACDiT}} = 4h \cdot \frac{L^2 + LB}{2} = 2hL^2 + 2hLB$$

Including projection and FFN cost, the total FLOPS under ACDiT is:

$$F_{\text{ACDiT}} = 2L\theta + 2hL^2 + 2hLB$$

**FLOPS Savings.**    The absolute FLOPS saved per layer is:

$$F_{\text{saved}} = F_{\text{full}} - F_{\text{ACDiT}} = 2hL^2(1 - \tfrac{B}{L})$$

The relative savings are:
$$\frac{F_{\text{saved}}}{F_{\text{full}}} = \frac{2hL^2(1 - \frac{B}{L})}{2L\theta + 4hL^2} = \frac{1 - \frac{B}{L}}{2 + \frac{\theta}{hL}}$$

When $L \gg h$, the savings approaches $\frac{1}{2}(1 - \frac{B}{L})$, implying up to 50% FLOPS reduction when $B \ll L$.

In practice, it is not always beneficial for small $B$. Setting $B$ to an excessively small value may not fully leverage the iterative modification inherent in the diffusion process, potentially compromising generation quality. Furthermore, given the parallel computing nature of computational kernels, a very small $B$ may not yield speed improvements, a phenomenon analogous to the rationale behind speculative decoding (Leviathan et al., 2023). Conversely, setting $B$ to a very large value diminishes efficiency both in terms of attention calculation. It also fails to capitalize on the strengths of auto-regressive generation. Indeed, when $B$ is set equal to $L$, ACDiT reverts to the original DiT model.

## B    Details of Model Architecture and Implementation

ACDiT mainly inherits the main architecture of DiT. We replace the original bidirectional attention with the proposed SCAM attention pattern to process the clean and noisy latent. Since we want to keep the architecture as simple and unified as possible, we use linear layers instead of convolution in the input layer and final layer. Besides, we replace the position embedding and Layer Normalization with RoPE (Su et al.,

2024) and RMSNorm (Root Mean Square Layer Normalization) (Touvron et al., 2023a), respectively. We find that QK-norm is important to stabilizing the video generation training, thus we use QK-norms in all experiments. The additional conditional information, including timesteps and labels, is injected into the model with AdaLN-Zero only on the noise part. For both image and video generation, we follow DiT and leverage the pre-trained image VAE (Kingma, 2013) from Stable Diffusion (Rombach et al., 2022), whose downsample factor is 8. For image generation under B block size, we group square latent representation patches with $\sqrt{B} \times \sqrt{B}$ shape as a block. We do not further train the tokenizer in the target dataset since it's out of the scope of our work, despite that further training might yield better results.

In the image generation task, we set the patch size as 1 and the autoregressive unit block size as $256 = 16 \times 16$. Therefore, for a $256 \times 256 \times 3$ image in $32 \times 32 \times 4$ latent shape, the total sequence length and autoregressive length are 1024 and 4, respectively. We explore 4 different model sizes, as shown in Table 5. ACDiT-B is used for design verification and analysis. ACDiT is trained on ImageNet for 1.2M iterations with a batch size of 1024. We use the AdamW optimizer (Loshchilov, 2017) and WSD (Warmup Steady Decay) learning rate scheduling (Hu et al., 2024) with the peak learning rate 3e-4 and no weight decay. The learning rate begins to decay in the last 15% training iterations. Following the common training recipe of generative models, we keep an exponential moving average (EMA) of the ACDiT weights during training using a decay rate of 0.9999. We sample images with DPM-Solver (Lu et al., 2022) for 25 steps within each block and use classifier-free guidance (Ho & Salimans, 2022) with a guidance scale of 1.5. In video generation, we sample 16 frames from each video and set the patch size as 2 and the block size as $1024 = 256 \times 4$. For a $16 \times 256 \times 256 \times 3$ video in $16 \times 32 \times 32 \times 4$ latent shape, the sequence length of each frame is 256 and the total sequence length is 4096, with 4 frames grouped into one block. We train ACDiT on UCF-101 for 400K iterations with a batch size of 96. The classifier-free guidance scale is 2.5. Other training configs are the same as image training. All models are implemented with PyTorch (Paszke et al., 2019). Specifically, we use FlexAttention[3] to implement the SCAM for both customization and efficiency. We use 256 and 96 NVIDIA H100 GPUs to train the image and video generation model, respectively.

## C  ACDiT PyTorch Code Example

In this section, we provide key implementation details of ACDiT, including (1) how clean and noisy blocks are arranged in the sequence, (2) how the Skip-Causal Attention Mask (SCAM) is constructed, and (3) how SCAM integrates with the attention module and KV-cache for efficient autoregressive inference.

For $N$ autoregressive blocks, we build a sequence of length $2N$ containing all clean blocks followed by all noisy blocks:

$$\texttt{idx\_clean}(i) = i, \quad \texttt{idx\_noisy}(i) = N + i, \quad i = 0, \dots, N - 1.$$

The following code shows how to construct SCAM, where noisy blocks are only allowed to attend to all previous clean blocks and themselves, while clean blocks attend to all preceding clean blocks. This attention mask enforces clean-past conditioning and enables KV-cache reuse across blocks.

```python
def build_attention_mask(self, T, B, device):
    size = T * B
    m_noise_noise = torch.zeros(size, size)
    for i in range(T):
        start_idx = i * B
        end_idx = start_idx + B
        m_noise_noise[start_idx:end_idx, start_idx:end_idx] = torch.ones(B, B)
    m_noise_clean = torch.zeros(size, size)
    for i in range(T):
        for j in range(i + 1, T):
            start_col = i * B
            end_col = start_col + B
            start_row = j * B
            end_row = start_row + B
```

---

[3]https://pytorch.org/blog/flexattention

```
15            m_noise_clean[start_row:end_row, start_col:end_col] = 1
16       m_clean_noise = torch.zeros(size, size)
17       m_clean_clean = torch.zeros(size, size)
18       for i in range(T):
19           start_idx = i * B
20           end_idx = start_idx + B
21           m_clean_clean[start_idx:end_idx, :end_idx] = 1
22
23       attn_mask = torch.zeros(2 * size, 2 * size)
24       attn_mask[:size, :size] = m_clean_clean
25       attn_mask[:size, size:] = m_clean_noise
26       attn_mask[size:, :size] = m_noise_clean
27       attn_mask[size:, size:] = m_noise_noise
28       return attn_mask.bool().to(device)
```

If using PyTorch FlexAttention, the same causal dependency can be implemented through its index-based mask generation. Below is the corresponding function:

```
1  def skip_causal_attn_mask_mod_gen(b, h, q_idx, kv_idx, block_size, len1):
2      q_idx = q_idx // block_size
3      kv_idx = kv_idx // block_size
4      mask = torch.where(((q_idx < len1) & (kv_idx < len1) & (q_idx >= kv_idx))| ((q_idx >= len1) &
       (kv_idx < len1) & ((q_idx - len1) > kv_idx)) | (q_idx == kv_idx) , True, False)
5      return mask
```

Once the mask is constructed, it is passed to the attention module. KV-cache is applied only to clean blocks, allowing us to cache context features while discarding the noisy-block activations after each diffusion step.

```
1  class SkipCausalAttention(Attention):
2      def forward(self, x, position_ids=None, attention_mask=None, block_size=None, cache=False):
3          B, N, C = x.shape
4          qkv = self.qkv(x).reshape(B, N, 3, self.num_heads, self.head_dim).permute(2, 0, 3, 1, 4)
5          q, k, v = qkv.unbind(0)
6          q, k = self.q_norm(q), self.k_norm(k)
7          if self.rope is not None:
8              q, k = self.rope(q, k, position_ids)
9          if self.caching:
10             if cache:
11                 if self.cached_k is None:
12                     self.cached_k = k[:, :, :block_size, :]
13                     self.cached_v = v[:, :, :block_size, :]
14                     self.cached_x = x
15                 else:
16                     self.cached_k = torch.cat((self.cached_k, k[:, :, :block_size, :]), dim=2)
17                     self.cached_v = torch.cat((self.cached_v, v[:, :, :block_size, :]), dim=2)
18             if self.cached_k is not None:
19                 k = torch.cat((self.cached_k, k[:, :, -block_size:, :]), dim=2)
20                 v = torch.cat((self.cached_v, v[:, :, -block_size:, :]), dim=2)
21         if not USE_FLEX_ATTENTION:
22             x = torch.nn.functional.scaled_dot_product_attention(
23                 q, k, v, attn_mask=attention_mask, dropout_p=self.attn_drop.p
24             )
25         else:
26             x = flex_attention(q, k, v, block_mask=attention_mask)
27         x = x.transpose(1, 2).reshape(B, N, C)
28         x = self.proj(x)
29         x = self.proj_drop(x)
30         return x
```

ACDIT uses a standard Transformer block design as in DiT, with one key modification: only the noisy latent tokens interact with the conditional information (diffusion timestep, class label). Clean latents remain unconditioned to preserve their deterministic role as the cached context.

```python
class ACDiTBlock(nn.Module):
    def forward(self, x, c, attention_mask=None, cond_length=0, block_size=None, cache=False,
    position_ids=None):
        N, T, _, C = c.shape
        N, TB, C = x[:, cond_length:].shape
        B = TB // T

        ada_c_list = self.adaLN_modulation(c).chunk(6, dim=-1)
        (shift_msa, scale_msa, gate_msa, shift_mlp, scale_mlp, gate_mlp) = [ada_c.repeat(1, 1, B,
    1).reshape(N, TB, C)for ada_c in ada_c_list]
        norm_x1 = self.norm1(x.to(torch.float32)).to(dtype)
        attn_input_x = torch.cat((norm_x1[:, :cond_length], modulate(norm_x1[:, cond_length:],
    shift_msa, scale_msa))dim=1)
        attn_output_x = self.attn(attn_input_x, attention_mask=attention_mask,
    block_size=block_size, cache=cache, position_ids=position_ids)
        x = x + torch.cat((attn_output_x[:, :cond_length], gate_msa * attn_output_x[:,
    cond_length:]), dim=1)

        norm_x2 = self.norm2(x.to(torch.float32)).to(dtype)
        gate_input_x = torch.cat((norm_x2[:, :cond_length], modulate(norm_x2[:, cond_length:],
    shift_mlp, scale_mlp)), dim=1)
        gate_output_x = self.mlp(gate_input_x)
        x = x + torch.cat((gate_output_x[:, :cond_length], gate_mlp * gate_output_x[:,
    cond_length:]), dim=1)
        return x
```

## D  Broader Impact

This work advances the efficiency and scalability of generative models for images, videos, and text by combining autoregressive modeling with diffusion. While such models enable positive applications in creative content generation, simulation, and representation learning, they may also be misused to generate misleading or harmful visual or textual content, including deepfakes. As with other general-purpose generative models, responsible deployment requires appropriate safeguards, dataset curation, and usage policies. We encourage future work to explore detection, watermarking, and alignment techniques alongside improvements in generative quality and efficiency.

## E  Qualitative Results of ACDiT

We show the qualitative results of ACDIT in Figure 7.

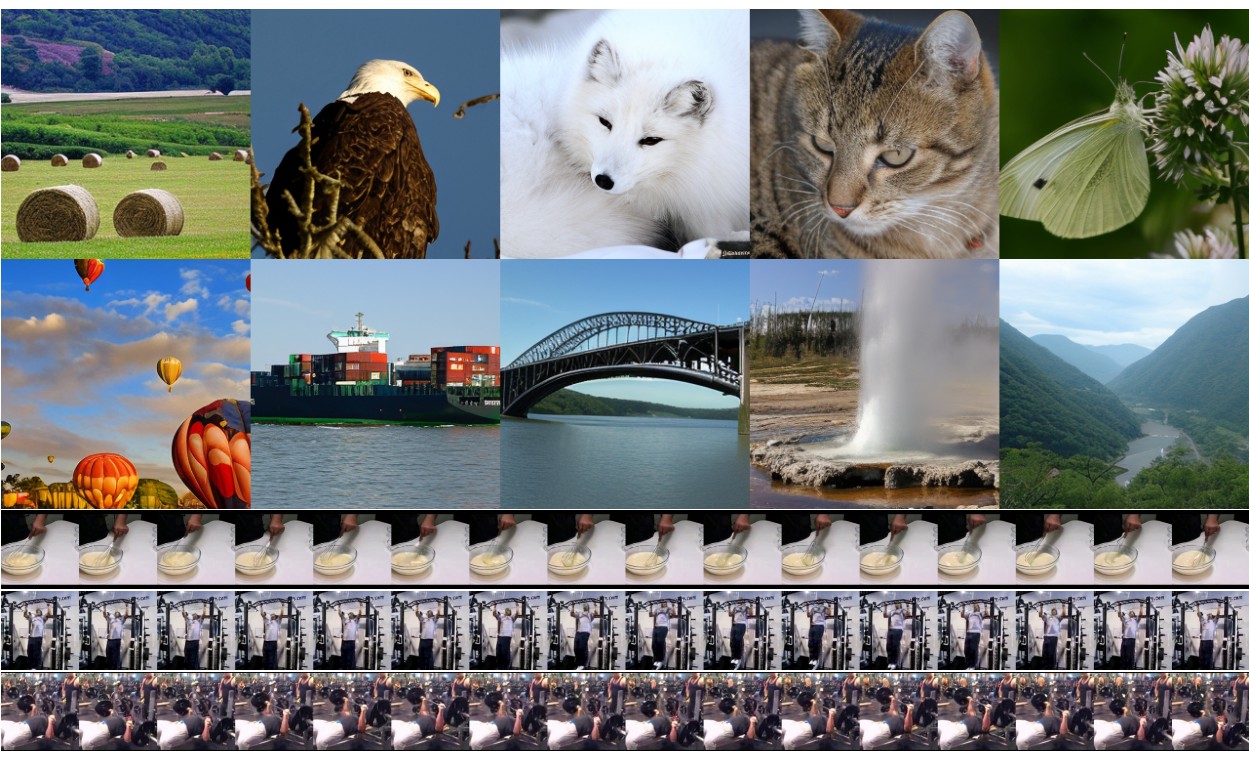

Figure 7: Sample images from ACDIT-H on ImageNet 256×256 and sample videos from ACDIT-XL trained on UCF-101.

