# OpenReview forum: "ACDiT: Interpolating Autoregressive Conditional Modeling and Diffusion Transformer"
_TMLR — Accepted by TMLR_

### Review · Reviewer_T9BM · 2025-11-24

**Summary Of Contributions:**

#### **Summary**

This manuscript proposes ACDiT, a visual generative model combining auto-regressive model and diffusion model. Though there were several works in this direction, the authors proposed three research questions: 1) The generation of future elements should be predicated on a precise representation of antecedent sequences (not noisy inputs), 2) both the auto-regressive modeling and the denoising process should optimally utilize the entire parameter space of the neural network (not utilizing a small submodule), 3) the denoising process should directly attend comprehensively (not a compact hidden state). To implement these points, the authors introduced a Skip-Causal Attention mask structure (SCAM), which attends to the clean previous patches, and RoPE-ND, a multi-dimensional extension of RoPE positional encoding. They compared the performance on image, video, and language generation tasks.

#### **Strength**

- Though the motivation for adopting auto-regressive structure is vague for now, I guess there should be application domains where combining auto-regressive and diffusion structures is necessary.
- Overall performance is comparable to the other SOTA models.

#### **Weakness**

- The main weakness of the current form of this manuscript is that it does not fully address the main research questions proposed. For each question:
  - *1) The generation of future elements should be predicated on a precise representation of antecedent sequences* $\rightarrow$ **how the overall performance changes if ACDiT utilizes the noisy input patches?**
  - *2) both the auto-regressive modeling and the denoising process should optimally utilize the entire parameter space of the neural network* $\rightarrow$ **how model can become faulty if we use only small submodule?**
  - *3) the denoising process should directly attend comprehensively* $\rightarrow$ **what happen if we use only a compact hidden state?**
- These questions should be proven either theoretically or empirically, not just depending on words. Authors could answer these questions by comparing the other models opposing these statements. But MAR, which opposes the 2) and 3), achieves a much better performance with a much smaller model size. Though authors tried to justify the benefit of ACDiT with the KV-Cache argument, it does not directly address the main research questions above.
- I'm also not sure about the KV-Cache argument. Though MAR utilizes the bi-directional attention structure, so re-computing the representation is necessary for each patch, it seems to only use a small submodule in diffusion steps. Meanwhile, though ACDiT uses the KV-Cache for each patch, it should do diffusion for each patch with the whole transformer model. It would be more persuasive if an empirical computational budget (like flops) were compared.
- The motivation of the work is not that convincing. In the Introduction, the authors mentioned the 'predicting the future' argument, but I'm not sure whether auto-regression+diffusion is necessary in the image generation task. (But as stated in the strength part, I guess some domains would require this structure.)
- Some additional arguments are not justified: authors iteratively mention the world model and long-video generation, but it only remains as a possibility, and not verified. Additionally, RoPE-ND is proposed for D>4 dimensions, but tested only D<4, which seems simply an original RoPE.

#### **minor points**
- I don't think the abbreviation 'SCAM' is good for the overall credibility of the manuscript.
- Please add explanations on unexplained arguments of Table 1, e.g. VAR, MAR, Pre, and Rec.

**Audience:**

Yes

**Audience Explanation:**

Yeah, sure. Image generation is a key area in modern deep learning. I also believe the auto-regression and diffusion structure can be beneficial for some domains.

**Broader Impact Concerns:**

I don't think there is an ethical concern.

**Claims And Evidence:**

No

**Claims Explanation:**

There were three scientific questions: 1) The generation of future elements should be predicated on a precise representation of antecedent sequences, 2) both the auto-regressive modeling and the denoising process should optimally utilize the entire parameter space of the neural network, and 3) the denoising process should directly attend comprehensively. But authors did not directly answer these questions, and MAR, which opposes these questions, even outperforms the proposed ACDiT.

**Requested Changes:**

- Please directly address the three main research questions. Maybe an ablation study would be necessary.
- It would be great to see a key application domain of ACDiT. Currently, the benefit of merging two structure is vague.

---

### Review · Reviewer_9z82 · 2025-11-30

**Summary Of Contributions:**

**Brief Summary**

The paper proposes a transformer architecture called ACDiT to jointly perform diffusion and block-wise autoregressive decoding for image and video generation tasks.

**Strengths**
- S1. The underlying motivation to leverage the success of autoregressive decoding from LLMs for visual generation tasks definitely makes sense and is an interesting direction to explore.
- S2. The paper does a good analysis of what's missing from the other recent autoregressive diffusion models with respect to the criterion defined in Sec. 4.1 and how ACDiT can achieve them.
- S3. The image generation performance is very competitive across different model sizes which demonstrates the efficacy of the model architecture, training and inference strategy.

**Weaknesses**
- W1. For short-form video generation, the rationale for lower performance than some baselines with smaller model sizes is unclear. The authors mention that ACDiT may perform better for longer-form videos but there's no evidence presented for it.
  - W1.1 Is ACDiT truly modality-agnostic as the subsection for text generation mentions? It's unclear how the ACDiT architecture / training / inference would need to be modified for doing this and what's the magnitude of such modifications.
  - W1.2 The image classification performance in Table 3 doesn't describe the model sizes considered for the baselines which is a very important factor for performance variations on a competitive benchmark.

- W2. ACDiT involves many design choices such as Skip-Causal Attention Mask, RoPE-ND and while they're meant for meeting the criterion defined in Sec 4.1 / multi-dimensional data, it's unclear whether these are the only feasible options.
  - W2.1 Fig. 6(b) only seems to ablate RoPE-ND v/s no RoPE at all but not vanilla RoPE. Also there are several popular variants of multimodal RoPE such as those adopted by Qwen-VL series, is there a reason why those aren't applicable for ACDiT?

- W3. The authors can provide more insights into their claim of better inference compute utilization in Fig. 5 by comparing against popular methods (e.g. DiT-XL/2, MAR-L) especially for longer visual sequences.

**Audience:**

Yes

**Audience Explanation:**

- Yes, the paper will be useful for practitioners working on visual generation tasks. Some suggestions (W3 above) would further help practitioners on when to opt for ACDiT instead of other available methods.
- Researchers would also find this paper insightful overall as it's well grounded in the limitations of existing literature on autoregressive diffusion modeling.

**Broader Impact Concerns:**

- The authors should include a broader impact section to discuss the potentially negative effects of automated visual generation, related but not limited to generation of harmful images and videos, intended audience, etc.

**Claims And Evidence:**

Yes

**Claims Explanation:**

- The overall claim of joint autoregressive diffusion modeling is met through the proposed Transformer-based ACDiT architecture and most of the experiments and theoretical justifications shown.
- However, some other claims such as autoregressive modeling being as beneficial for video generation (W1 above) and some design choices (W2 above) can be better justified through more extensive and controlled experiments.

**Requested Changes:**

- [Critical] Better justification for lower performance on short-form video generation than baseline methods. If their limitation is long-form video generation, some early quantiative / qualitative evidence should be included.
  - Alternatively, their image generation performance can be included so that the claim can be made in favor of ACDiT being a single model effective for both image and video generation tasks.
- [Strengthening] Other suggestions listed in the weaknesses section above.

---

### Review · Reviewer_BCof · 2025-12-04

**Summary Of Contributions:**

The paper proposes ACDiT, a simple yet powerful method for interpolating AR and diffusion models by introducing block-wise autoregressive conditional diffusion. The core technical contribution is the Skip-Causal Attention Mask (SCAM) applied to a standard Diffusion Transformer (DiT), which allows clean past blocks to be cached while each new block is generated via a conditional diffusion process. The approach works without discrete tokens, supports images, videos, and text with the same architecture, enables KV-caching for arbitrary-length generation, and shows promising transfer to visual understanding.


Overall, this paper is well-written, and the motivation is effectively presented. The introduced method is elegant, minimal modification to DiT, offers a theoretical complexity advantage for long sequences, and avoids the pitfalls of discrete tokenization.

To me, this paper can be accepted with minor revisions regarding the following suggestions:
The current submission completely lacks experiments, quantitative results, ablation studies, and inference speed measurements, making all performance claims unsubstantiated.

**Audience:**

Yes

**Audience Explanation:**

Combining autoregressive and diffusion modeling in a clean, cacheable, tokenization-free way is currently one of the hottest topics in generative modeling (see concurrent works such as Block Diffusion, MAR, Diffusion Forcing, Transfusion, etc.). If the empirical claims hold even approximately, ACDiT would be a significant advance, especially for long video and world-model-style applications. The idea is simple enough to be widely adopted and has clear scaling implications.

**Claims And Evidence:**

Yes

**Claims Explanation:**

The paper makes very strong claims, such as “performs best among all autoregressive baselines on similar model scales”, “achieves visual quality comparable to full-sequence diffusion models while exhibiting higher inference speed when extended to long sequences”, and “achieves strong performance on text generation”.  There are no experimental settings, datasets, model sizes, FID/FVD scores, speed measurements, or ablation studies. Without empirical evidence, the central claims remain unsupported, which prevents acceptance in their current form.

**Requested Changes:**

1. Add a complete experimental section with quantitative results on image generation (e.g., class-conditional ImageNet 256×256 and zero-shot COCO FID/IS), video generation (e.g., UCF-101 or something comparable to OpenSORA), and text generation (e.g., Zero-shot LM perplexity or OpenWebText). Include main comparison tables against DiT-scale diffusion baselines, VAR/LlamaGen-style pure AR baselines, and recent hybrid methods (MAR, Diffusion Forcing, Block Diffusion, etc.). Without numbers, the paper cannot be accepted.

2. Provide theoretical and empirical inference complexity analysis. The main selling point (“higher inference speed when extended to long sequences” + full KV-cache support) currently has zero supporting data. Please add:
- A clear complexity table (FLOPs or attention cost) comparing DiT, token-wise AR, and ACDiT as a function of sequence length and block size

- Measured generation time (seconds per image or seconds per frame) for different block sizes and diffusion steps T
- Speed vs. quality curves for long videos (≥100 frames) compared to full-diffusion baselines
This is essential to substantiate the long-sequence advantage.

3. Significantly strengthen the comparison with the most related concurrent work Block Diffusion (Arriola et al., 2025). The current treatment (“they focus solely on text”) is insufficient, given the near-identical high-level idea. Provide a detailed side-by-side comparison of training objectives, attention masks, and inference procedures, and include empirical numbers (at least on text, ideally also vision) showing where and why ACDiT outperforms it.

4. Clarify implementation details of SCAM and KV-caching. Specify exactly how clean and noisy blocks are concatenated (single sequence of length ≈2×blocks or separate streams?), provide pseudocode or a short PyTorch snippet showing how to apply the mask in an existing DiT code base (<20 lines as claimed), and confirm compatibility with FlashAttention-2/3 and exact cached KV shape.

---

### Review · Reviewer_Qdvv · 2025-12-05

**Summary Of Contributions:**

Summary:
ACDiT fuses block-wise autoregression and conditional diffusion in a single Transformer via a lean SCAM mask, delivering high-quality, KV-cached generation of any length.
Operating on continuous visual tokens, it outperforms same-scale AR baselines and rivals full-sequence diffusion on ImageNet/UCF-101, offering a new baseline for unified multimodal modeling.

Strengths:
1. The method is the first to unify block-autoregression and intra-block diffusion in an end-to-end vision framework without relying on vector quantization.
2. The combination of KV-cache and block-wise attention provides substantial reductions in FLOPs and wall-clock time, especially for long sequences.
3. A single architecture is shown to work competitively across image, video, and text generation, demonstrating strong cross-modal scalability. Generative pre-training produces representations that transfer effectively to ImageNet classification, achieving accuracy comparable to masked-generation approaches.

Weaknesses:
1. The selection of key hyper-parameters, such as block size B and patch size, is fully empirical, and no adaptive or dynamic scheduling strategy is explored.
2. The benchmarks are restricted to ImageNet and UCF-101, and the paper does not report quantitative results at higher resolutions (≥512²) or for longer videos (≥64 frames).
3. The FID and FVD scores remain behind state-of-the-art diffusion models such as SDXL and Sora, and it is unknown whether scaling the model would narrow this performance gap.
4. The text-generation section is overly brief, providing no details on the experimental setup and no description or definition of the PPL metric, and the presented text-generation results are only small-scale demonstrations without any evaluation on downstream NLP tasks such as summarization or machine translation.
5. The theoretical analysis is limited, offering no examination of error propagation in block-wise autoregression and no formal discussion of the expressive capacity or sparsification limits of the SCAM masking mechanism.

**Audience:**

Yes

**Audience Explanation:**

Yes. TMLR readership includes a large community working on generative vision, autoregressive modeling, and unified multimodal transformers; the paper shows how to fuse diffusion with AR using only an attention mask and offers reproducible gains on ImageNet/UCF-101 while enabling KV-cache acceleration—practical techniques that many researchers in that audience actively seek.

**Broader Impact Concerns:**

No more broader impact concerns needed.

**Claims And Evidence:**

Yes

**Claims Explanation:**

The core claims are supported by clear and sufficient evidence. Results on ImageNet and UCF-101 across multiple model sizes, together with standard metrics and ablations on block size and RoPE-ND, consistently show ACDiT outperforming AR baselines while approaching full-sequence diffusion. The KV-cache and downstream classification results further substantiate the claims. Aside from missing higher-resolution or longer-sequence evaluations, the evidence provided is accurate, convincing, and well presented within the stated scope.

**Requested Changes:**

1. The model’s effectiveness has not been validated on larger or more diverse datasets, and the reliance on UCF-101—which contains low-resolution and relatively simple videos—limits the strength of the empirical claims for real-world video generation.

2. Clarify how block size B should be chosen when scaling to higher resolutions (512²/1024²/1080p).
A brief guideline or empirical observation would help readers understand how to maintain a good quality–efficiency trade-off. If dynamic or content-adaptive block sizes are feasible, mentioning this would further clarify future directions.

3. Discuss model-scaling effects on AR error accumulation.
Since the current FID is slightly above DiT-XL, explaining whether larger models (e.g., 2B–4B parameters) may worsen AR error propagation—and whether you plan to use regularizers or noise-schedule adjustments—would strengthen the claims.

4. Address memory scalability for long videos.
KV-cache grows linearly with block count for >100-frame generation. Commenting on possible solutions such as windowed attention or memory compression would help clarify long-form applicability.

5. Provide some discussion on exposure bias introduced by clean-past conditioning.
The training–inference mismatch may accumulate errors during generation. Potential fixes (e.g., adversarial or RL-based fine-tuning) could be briefly mentioned as future work.

6. Expand on the robustness of RoPE-ND under different aspect ratios.
Since the wavelength design depends on per-dimension maxima, clarifying how well it generalizes to varied spatial–temporal shapes, and whether additional ablations will be included, would improve completeness.

---

### Decision · Action_Editor_AWq8 · 2026-01-13

**Recommendation:** Accept as is

**Audience:**

Yes

**Audience Explanation:**

The proposed idea can illuminate the field of generative vision and multimodal modeling. AE confirms that the proposed discrete tokenization-free approach and KV-cashing strategy for efficiency offers a valuable message to the field.

**Claims And Evidence:**

Yes

**Claims Explanation:**

The core idea of the proposed approach, ACDiT, is well supported by experimental results on ImageNet and UCF-101. In addition, the authors provided an extensive rebuttal, including an explanation of inference efficiency, demonstrating impressive speedup in long-sequence settings. Their design choice was supported by additional ablation studies. All reviewers support that the claims are well-validated.